# *HOXA9* has the hallmarks of a biological switch with implications in blood cancers

Laure Talarmain[1], Matthew A. Clarke[2], David Shorthouse[3], Lilia Cabrera-Cosme[4], David G. Kent[4], Jasmin Fisher[2] & Benjamin A. Hall[3]

Blood malignancies arise from the dysregulation of haematopoiesis. The type of blood cell and the specific order of oncogenic events initiating abnormal growth ultimately determine the cancer subtype and subsequent clinical outcome. HOXA9 plays an important role in acute myeloid leukaemia (AML) prognosis by promoting blood cell expansion and altering differentiation; however, the function of HOXA9 in other blood malignancies is still unclear. Here, we highlight the biological switch and prognosis marker properties of HOXA9 in AML and chronic myeloproliferative neoplasms (MPN). First, we establish the ability of HOXA9 to stratify AML patients with distinct cellular and clinical outcomes. Then, through the use of a computational network model of MPN, we show that the self-activation of HOXA9 and its relationship to JAK2 and TET2 can explain the branching progression of JAK2/TET2 mutant MPN patients towards divergent clinical characteristics. Finally, we predict a connection between the RUNX1 and MYB genes and a suppressive role for the NOTCH pathway in MPN diseases.

Blood cancers are malignancies that can arise from any type of blood cell and dramatically affect haematopoiesis. Myeloproliferative neoplasms (MPNs) are chronic diseases of the myeloid lineage characterised by an excessive production of fully functional terminally differentiated blood cells. These have been classified into three types: polycythemia vera (PV), essential thrombocythemia (ET), and primary myelofibrosis (PMF)[1]. Despite the relatively good prognosis of these diseases, MPN patients are at high risk of thrombosis and can develop a blast phase MPN (MPN-BP)[2]; a subtype of the blood cancer acute myeloid leukaemia (AML) with poor survival outcomes[3]. The frequency of MPN transformation to MPN-BP is highly related to the initial MPN disease type[4–6]. Therefore, a better understanding of the molecular events driving the different subtypes of MPNs is essential to help diagnose patients with higher risk of thrombosis and AML progression.

AML itself is an aggressive blood and bone marrow malignancy defined by the uncontrolled growth of myeloid progenitor cells along with a myeloid-lineage differentiation arrest[7]. As with MPN, there exist different types of AML with a broad range of morphologic, cytogenic, and immunologic features, all associated with diverse clinical outcomes[8]. Despite their similarities, prognosis, symptoms, and genetic alterations differ between AML and MPN. For example, *JAK2* mutation is the main driver event of MPN diseases yet is rarely found in de novo AML[9]. However, myeloid-lineage dysregulation occurs in both MPN and AML, and alongside the ability of MPN to evolve to AML, this may suggest that both diseases share common biological mechanisms. The identification of these processes could help identify aberrant genes and pathways involved in both AML and MPN to detect MPN patients with higher risk of developing AML.

Better understanding of the patterns of genetic alterations in cancer cells can be used for the classification of blood diseases and prediction of progression into more severe forms of the disease[10]. How different combinations and orders of mutations lead to different

[1]Peter MacCallum Cancer Centre, 305 Grattan Street, Melbourne, VIC 3000, Australia. [2]UCL Cancer Institute, University College London, Paul O'Gorman Building, 72 Huntley Street, London WC1E 6BT, United Kingdom. [3]Department of Medical Physics and Biomedical Engineering, Malet Place Engineering Building, University College London, Gower Street, London WC1E 6BT, United Kingdom. [4]York Biomedical Research Institute, Department of Biology, University of York, York YO10 5DD, United Kingdom. ✉e-mail: b.hall@ucl.ac.uk

subtypes of cancer remains a major open question[11,12]. The importance of mutation order has been demonstrated in MPN by Ortmann et al.[13], who show that two subpopulations of patients with MPN can be distinguished by the order of mutation acquisition between the *TET2* and *JAK2* genes, and that these subpopulations have distinct clinical characteristics. Further analyses of these cohorts show that patients with *JAK2* mutated before *TET2* are younger at presentation of the disease in clinics, are more likely to present with PV, have a higher risk of thrombosis, and respond better to JAK2 inhibitor ruxolitinib. However, the molecular interplay between both mutations within cancer cells and how their order rather than their combination triggers dissimilar clinical characteristics have not been investigated.

Overexpression of a single homeobox gene, *HOXA9,* has been reported as sufficient to quickly induce myeloproliferation, gradually followed by AML progression after a period of time[14]. Homeobox genes or HOX genes were first identified in the fruit fly *Drosophila melanogaster* as essential regulators of early embryogenesis[15] and are thought to have a critical role in cancer development[16]. In the HOXA family, *HOXA9* is the most described gene in literature and its expression was shown to be the single most highly correlating factor (out of 6817 genes tested) for poor prognosis in AML[17]. The importance of *HOXA9* in AML has been widely explored; however, this has mainly focused on specific AML subtypes such as MLL-rearranged leukaemia[18] and NUP98-*HOXA9* induced leukaemia[19], while its role in other blood malignancies such as MPN or other AML subtypes is poorly characterised. Recently, the oncogenic property of *HOXA9* has been associated with its self-positive feedback loop in myeloid precursor cells as a result of its ability to bind its own promoter[20]. We hypothesise in this work that, as a consequence of the underlying gene network, the expression of *HOXA9* could be used to stratify patients according to risk with blood cancers affecting the myeloid lineage.

In this study, we define a switch as a molecule that has the ability to self-sustain a positive feedback loop. Using public datasets from AML patients and MPN studies, we show that bimodal *HOXA9* expression identifies two distinct cohorts of patients/mice, reflecting the gene acting as a binary switch in the cell. Firstly, *HOXA9* bimodal expression in AML is associated with clinical features, such as age and WBC counts, but also patient classification into specific French-American-British (FAB) or molecular subtypes. Secondly, we design a computational network model that offers a mechanistic explanation of the distinct clinical features of MPN progression in patients with different orders of *JAK2* and *TET2* mutations (Fig. 1). Using our computational model and experimental validation, we argue that *HOXA9* is downstream of *JAK2* and *TET2* and effectively stores their mutational history. This "memory" property of HOXA9 is induced by the presence of its self-activation, captured by a positive feedback loop in our

model. This results in a phenotypic switch in double mutant cells with different mutation orders producing distinct subtypes of the disease. Finally, the network model also predicts a suppressive role for the NOTCH pathway in MPN and an interaction between *RUNX1* and *MYB*.

## Results

### *HOXA9* expression separates cohorts of AML patients with distinct clinical features

Ectopic expression of *HOXA9* in AML has been widely demonstrated, but few studies have investigated the biological attributes of this transcription factor contributing to leukaemogenesis. Zhong et al.[20] have shown that *HOXA9* in cell lines can induce its own expression through a positive feedback loop, which promotes a continuous differentiation block and self-renewal leading to increase of hematopoietic stem cells and development of leukaemia. To validate *HOXA9* self-activation and the oncogenic role in leukaemia in patients, we studied its expression in untreated de novo AML RNA sequencing data from The Cancer Genome Atlas (TCGA)[21]. We find that *HOXA9* has bimodal expression in this data set (Fig. 2a). Whilst we find other *HOX* genes to also have a bimodal expression, they are correlated or anti-correlated with *HOXA9* and to the best of our knowledge *HOXA9* is uniquely downstream of both *JAK2* and *TET2*. We further find two other genes with bimodal expression, *APP* and *IGSF10*, which are not clearly correlated to *HOXA9* status. Correlation or anti-correlation with *HOXA9* confounds survival analysis, limiting our ability to analyse the contribution of the second gene. We do, however, find within low or high *HOXA9* cohorts no significant survival differences with *IGSF10*, and a survival difference between high/low *APP* expression within the cohort with low APP expression. To explore the differences between patients with different levels of *HOXA9* expression and disregard external factors that could cause this bimodality, we separated patients into two cohorts, with 31 patients in the low expression peak, and 80 patients in the high expression peak. A survival analyses of both groups using Kaplan–Meier survival curves and the log-rank test confirmed that *HOXA9* can be used as a marker of poor prognosis in AML ($p < 0.001$, Hazard Ratio 0.29 for low expression) (Fig. 2b) regardless of age (Fig. S3). This patient stratification based upon *HOXA9* expression is consistent with the reported positive feedback loop characteristic of this gene and suggests that once activated or inhibited, the gene would maintain its expression level, leading to divergence in the disease progression.

To investigate if the switch role property of *HOXA9* impacts AML subtypes, we looked at the distribution of FAB (named M0–M7) and molecular classifications among the two *HOXA9* cohorts. We show that different *HOXA9* expression cohorts exclude specific FAB subtypes (Fig. S4a). This suggests that *HOXA9* expression is strongly coupled to some FAB subtypes.

In light of these findings, we looked to characterise the common features of *HOXA9* expression cohorts. Cytogenic aberrations and gene rearrangements are frequent in AML and are known to alter the disease morphology as well as the clinical features and prognosis[21]. We found that *HOXA9* expression separates patients with different molecular classification (Fig. S4b). MLL-induced leukaemia has been linked to high *HOXA9*[18], while M3 AML subtype is characterised by *PML-RARα* translocation and low *HOXA9* in the literature[22]. Low *HOXA9* expression in AML with *RUNX1-RUNXT1* and *CBFB-MYH11* abnormalities, which constitute the core binding factor (CBF) AML, has also been established in literature[23]. Our findings confirm these observations and further establish the correlation between high *HOXA9* expression and the M0 and M5 subtypes in addition to complex cytogenetics. Finally, we searched for other clinical differences between cohorts, finding that *HOXA9* expression correlates with age, white blood cell count (WBC) and blast percentage in the bone marrow (Fig. S4c). These divergent characteristics between cohorts suggest that the observed bimodality is not induced by external/sequencing factors.

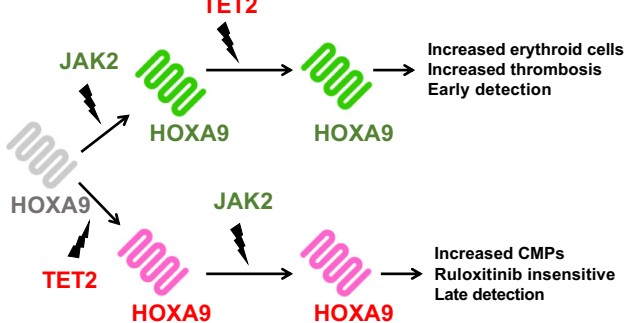

**Fig. 1 | Temporal mutation order has clinical implications in myeloproliferative neoplasms.** CMPs: common myeloid progenitors. Both of the genes JAK2 and TET2 are mutated in myeloproliferative neoplasms, but different orders of the mutation can lead to distinct changes in the cell types that expand, drug sensitivity, and time of presentation in the clinic.

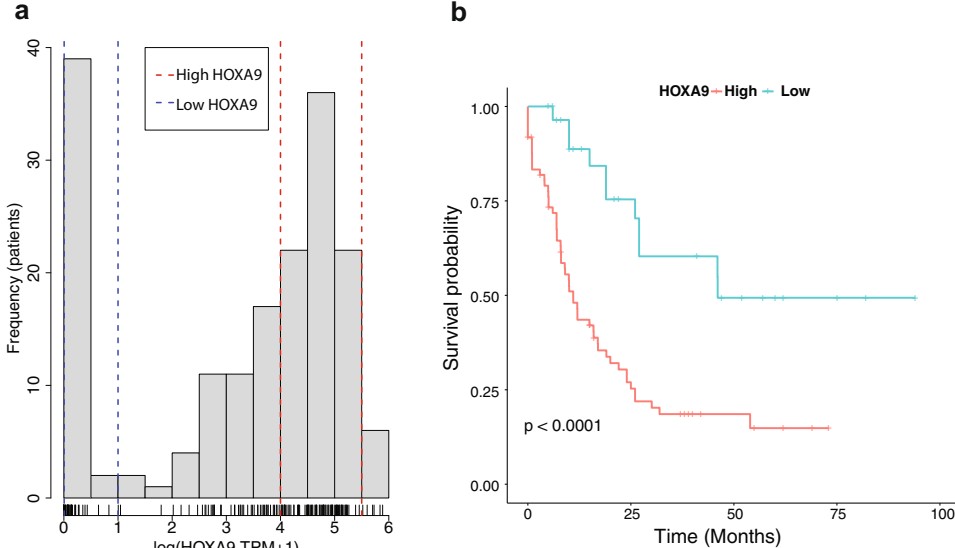

**Fig. 2 | *HOXA9* low and high expression stratifies patients in AML. a** *HOXA9* expression in AML patients is significantly bimodal (unimodality test rejected with $p < 2.2 \times 10^{-16}$), suggesting a role as a genetic switch. The low peak consists of 31 patients for *HOXA9* expression between 0.005–1 log (TPM+1) and 80 patients in the high peak −4–5.5 log(TPM + 1). **b** *HOXA9* high and low cohorts have divergent AML prognosis, consistent with known *HOXA9* biology in AML (log-rank test, $p = 6 \times 10^{-5}$). TPM transcript per million. Source data are provided as a Source Data file.

*PML-RARα*, *RUNX1-RUNXT1* (AML1-ETO), or *CBFB-MYH11* chromosomal abnormalities predict good prognosis in AML patients[24,25]. All these aberrations are linked to low *HOXA9* expression which also exhibits good survival prognosis among patients compared to high expression. To confirm that high *HOXA9* is a marker of poor prognosis independently of its associated molecular aberrations or FAB subtypes, we studied survival outcomes within FAB classes. As M0, M3, and M5 are exclusively in one cohort, we examined the survival of patients within the M2 and M4 subtypes for high and low *HOXA9* expression. Survival curves and log-rank tests within both subtypes confirms that high HOXA9 is a marker of poor prognosis *HOXA9* (Fig. S8 and S9). Overall our findings are consistent with *HOXA9* becoming trapped in high- or low- expression states through self-activation in AML diseases.

## The *JAK2*/*TET2*/*HOXA9* motif can explain divergent disease clinical outcomes in MPN

The identification of MPN patients at higher risk of developing AML remains a major clinical challenge. *JAK2* is the most commonly mutated gene in many MPN patients, but different subtypes of the disease with distinct clinical traits are observed[26]. In contrast, *TET2* was only recently identified in blood studies. First discovered in MPN in 2008 by Delhommeau et al.[27], *TET2* mutation resulting in its loss of function has been associated with diverse haematologic malignancies[28]. We have shown that *HOXA9* can enable clinical stratification in AML, potentially due to the presence of a positive feedback loop. Ortmann[13] describes a bifurcation among MPN patients that acquire *JAK2* and *TET2* mutations in different orders. This raises the question whether *HOXA9* expression could also explain these divergent clinical symptoms and help stratify MPN patients with low and high risk to develop AML.

To address this question, we constructed a computational network model in a multistep process. In order to reproduce the branching in MPN patients, the underlying network of gene interactions must include genes that are sensitive to the mutation order[29]. This requires that parts of the network act a switch, capable of storing "memory" of previous events. This "memory" property can be encoded by a positive feedback loop acting on a gene that is downstream of both mutated genes[30]. This hypothetical gene must additionally respond differently to each of the mutations. That is to say, one

mutation must activate the gene whilst the other reduces it, so that the gene can maintain its change in activity after the occurrence of the second mutation. The loop is necessary to induce this inheritable change in the presence of constitutive reset processes such as protein and RNA degradation.

We developed a computational model of this simple gene motif with *JAK2* and *TET2* genes and a hypothetical gene target with a positive feedback loop (Fig. 3a). *TET2* and *JAK2* have been indirectly and directly linked to *HOXA9* activity. *STAT5* is a well-known downstream target of *JAK2*[31], and it is also established that *STAT5* and *HOXA9* act as binding partners in hematopoietic cells[32]. Furthermore, it was recently shown that tyrosine phosphorylation of *HOXA9* is *JAK2*-dependent[33] and seems to increase the effect of *HOXA9* on its downstream targets[33]. Regarding the interaction of *TET2* with *HOXA9*, Bocker et al. found significantly reduced expression of *HOXA* genes when *TET2* expression is lost[34]. In particular *HOXA9* expression in kidney is significantly decreased by *TET2* loss. *HOXA9* is therefore activated by *JAK2* and reduced by *TET2* loss and possesses a self-positive feedback loop property[20]. Therefore, the *JAK2*/*TET2*/*HOXA9* motif shares all the required properties for observing a clinical divergence in blood diseases.

Based on this *JAK2*/*TET2*/*HOXA9* motif, we refined our computational model to reproduce the observed biological differences between patients with different combinations of *JAK2* and *TET2* mutations (Supplementary Data). To do so, we extended our computational network with six phenotypes relevant to cancer development: stem cell self-renewal, common myeloid progenitor (CMP) expansion, granulocyte-monocyte progenitor (GMP) expansion, GMP differentiation, erythroid differentiation and megakaryocyte-erythroid progenitor (MEP) expansion (Fig. 3b). We further included important hematopoietic markers in our computational model. We found that additional interactions such as the activation of *MYB* by *RUNX1* are also required to reproduce the correct biological features of MPN (Table 1). A detailed literature review and full description of how we built the network model are available in Tables S1–3 and the Supplemental Methods.

Finally, four fundamental cancer genotypes are defined: the wild-type (no mutation), the *TET2* single mutant, the *JAK2* single mutant and the double mutant (in either order) (Table 1). The wild-type model illustrates haematopoiesis in its healthy state. The single mutants are

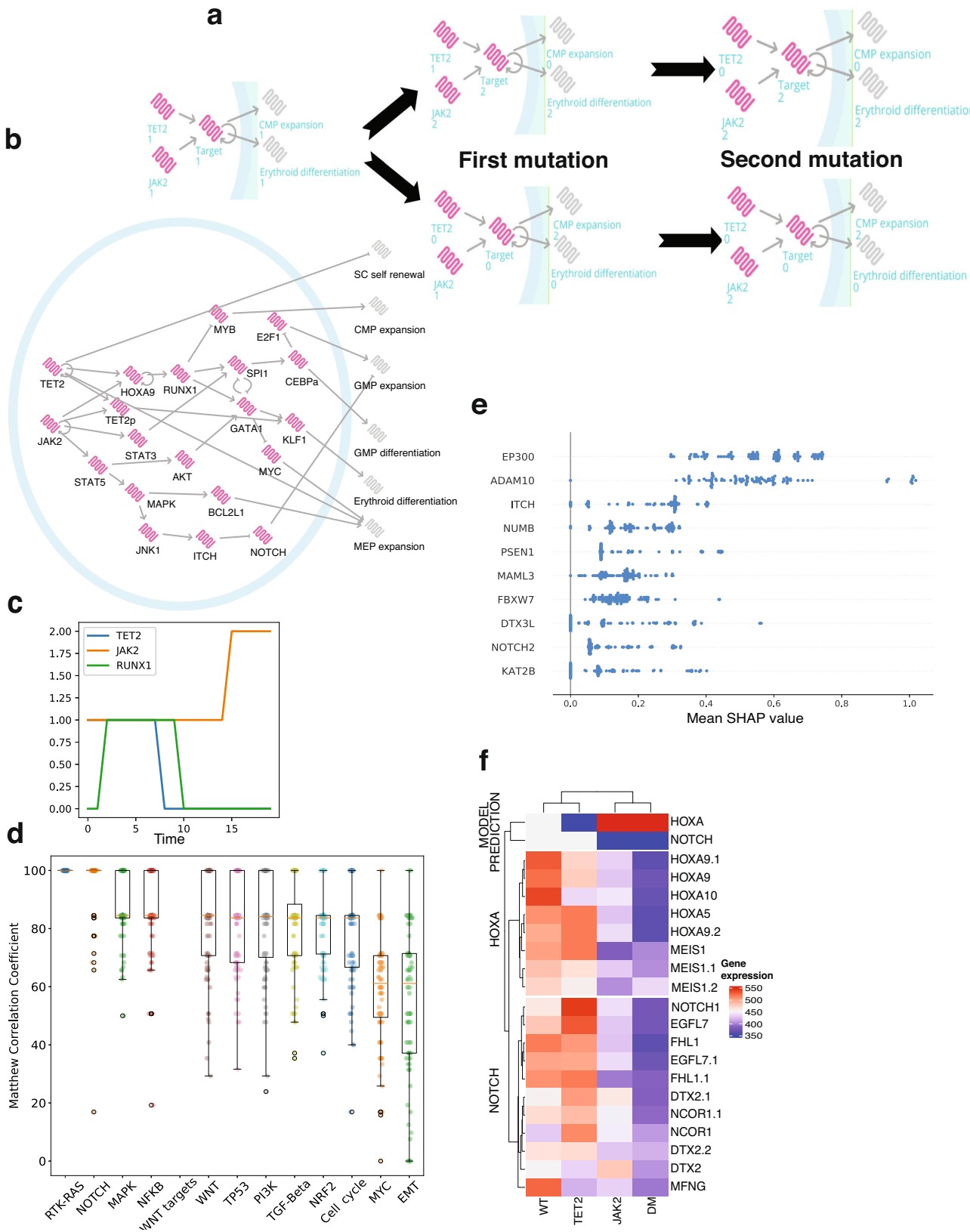

defined using the literature (see Supplementary Information). The final genotype is the double mutant which can lead to one of two cancer endpoints (fixpoint attractors that represent one of the two clinical outcomes). Each fixpoint represents either *TET2* first or *JAK2* first double mutants and are defined from results presented by Ortmann et al.[13]. Our computational model as shown in Fig. 3b reproduces the

expected behaviours described in Table 1 and therefore the clinical stratification observed in Ortmann et al.[13]. The model suggests that the elevated differentiation observed in the *JAK2* first double mutants[13] is induced by the increased expression of *RUNX1, KLF1* and *GATA1* as well as the downregulation of *MYC* not found in *TET2* first double mutant. This gene expression difference between double mutants can partly

**Fig. 3 | _HOXA9_ determines cancer cell fates in MPN diseases with _JAK2_ and _TET2_ mutations. a** MPN patients with JAK2/TET2 mutations showing different clinical characteristics can be explained by a simple gene motif including a switching property. The model starts from a healthy state on the left (wild-type) and sequentially acquires mutations in JAK2 and TET2 genes. The first mutation affects the gene target expression (middle networks) which remains stable when the second mutation appears. The order in which mutations occur impacts on the gene target expression but also the phenotypes, common myeloid progenitor (CMP) expansion, and erythroid differentiation (networks on the right). **b** The JAK2/TET2/HOXA9 molecular network is built with the BioModelAnalyzer (BMA) tool and integrates six output phenotypes: stem cell (SC) self-renewal, CMP expansion, granulocyte-monocyte progenitor (GMP) expansion, GMP differentiation, erythroid differentiation, and megakaryocyte-erythroid progenitor (MEP) expansion. The bifurcation analysis identifies two stable states in the double mutant with different phenotype values that fit the mutation order characteristics observed in MPN patients with JAK2 and TET2 mutations.

**c** BioModelAnalyzer simulation shows that RUNX1 expression is unchanged by JAK2 activation mutation following TET2 loss. **d** We select AML patients with lowest (30) and highest (30) JAK2 expression for the purpose of classification. Ranking of major cancer pathways determine the RTK-RAS pathway as the most correlatd to JAK2. This pathway contains JAK2 and so this result is as expected. The second pathway is NOTCH. The overall accuracy of the pathway is computed with the Matthews Correlation Coefficient. Box plots represent interquartile range and whiskers 1.5 × IQR. **e** We plot SHapley Additive exPlanations (SHAP) mean score of each model for the NOTCH pathway to determine which genes of the pathway have been important classifiers; that is, genes with an important expression correlation with JAK2. **f** The heatmap of the NOTCH pathway and HOXA family generated using MPN microarray datasets from[43] validate NOTCH expression in our model. HOXA heatmap confirms HOXA bimodality. "JAK2" and "TET2" refer to the single mutant mouse models, "DM" the double mutant with JAK2 mutated first and "WT" the wild-type (no mutation) genotype. Source data are provided as a Source Data file.

explain the divergent clinical behaviours between the two groups of patients, including the increased risk of thrombosis and the faster diagnosis as a result of the abnormally high number of differentiated cells in these patients.

The self-loop on _HOXA9_ plays a fundamental role in determining model behaviour. To explore how it influences cell phenotypes, we tested three different possible outcomes of removing it from the model. Simply removing _HOXA9_ self-activation in our model results in its stable overexpression in the double mutant genotype (Fig. S5), as the impact of the _JAK2_ mutation overwhelms the effect of the _TET2_ mutation. However, the loss of this interaction could lead to more complex outcomes. For example, _HOXA9_ may be dependent on a basal level of self-activation to act in the cell. To explore this, we also tested the case where removal of the self-loop causes _HOXA9_ null activity (Fig. S6) and stabilisation of the double mutant. Finally, the impact of the mutations could compensate for one another in the absence of the self-loop, leading the double mutant to have wild-type activity levels. In this situation, the model is unstable due to the interactions between _SPI1_ and _GATA1_, though fewer variables are involved in the instability (Fig. S7). We conclude generally that loss of _HOXA9_ self-activation leads to the partial or total loss of bifurcation in the model and its responsiveness to the order of mutations. This reinforces the importance of the self-positive feedback loop in determining cell phenotype and the subsequent clinical separation of patients with differing orders of _JAK2_ and _TET2_ mutations.

## MPN network predicts gene dynamics and interactions

The computational model identifies gene dynamics as part of the MPN disease progression. In the complete network model, _HOXA9_ requires both _JAK2_ and _TET2_ expression to remain active (Table S3).

**Table 1 | Specification table for _JAK2_/_TET2_ BMA model**

|  | WT | TET2 | JAK2 | TET2 first | JAK2 first |
|---|---|---|---|---|---|
| Stem cell renewal | 1 | 2 | 1 | 2 | 2 |
| CMP expansion | 1 | 2 | 1 | 2 | 1 |
| GMP expansion | 1 | 2 | 2 | 2 | 2 |
| GMP differentiation | 1 | 0 | 1 | 1 | 1 |
| Erythroid differentiation | 1 | 0 | 2 | 1 | 2 |
| MEP expansion | 1 | 1 | 2 | 2 | 2 |

These specifications are established phenotypic features and are used to test model correctness. In order from the left to the right columns, they are the wild-type state, the _TET2_ single mutant, the _JAK2_ single mutant and finally the double mutants, which consists of a bifurcation with two state attractors that represent the case where _TET2_ is mutated before _JAK2_ (_TET2_ first) and the alternative case where _JAK2_ is mutated first. We determine phenotype values using literature for the single mutants and Ortmann et al.[13] for the double mutants. The value 1 represents the healthy state, 0 the lowered/inactive state, and 2 the overactive state.

Upregulation of either _JAK2_ or _HOXA9_ results in the hyperactivation of _HOXA9_ while _TET2_ loss causes inactivation. Wild-type activity is maintained by the balance of these two genes. _JAK2_ activation mutation and _TET2_ loss both drive the system into a committed state. _JAK2_ activation raises _HOXA9_ activity to a level at which it can maintain its activity through control of its own expression. Subsequent loss of _TET2_ does not impact its activity as this hyperactivation makes it independent of _TET2_. Conversely, _TET2_ loss causes a loss of _HOXA9_ expression in the cell, rendering it insensitive to subsequent _JAK2_ activation. This occurs as _HOXA9_ expression drops, preventing a subsequent response to _JAK2_ activation due to low concentration of the protein in the cell. Therefore, a possible explanation of the order dependence could be a combination of mutual dependency between _JAK2_ and _TET2_ to activate _HOXA9_, combined with the positive feedback self-loop of _HOXA9_ itself.

One key feature of _TET2_-first MPN patients is their reduced sensitivity to Ruxolitinib, a JAK2 inhibitor drug[13]. Interestingly our computational model suggests that after _TET2_ loss, _RUNX1_ expression is unchanged by _JAK2_ activation mutation due to the "switching" property exerted by _HOXA9_ self-loop (Fig. 3c). However, this gene is affected by _JAK2_ mutation in the context of _TET2_ wild-type expression (Fig. S10). It follows that _JAK2_ inhibition is therefore inefficient for this important hematopoietic regulator which could explain the reduced effect of Ruxolitinib in _TET2_ first patients.

Whilst building single mutant phenotypes, we noticed a relationship between _JAK2_ and GMP expansion is required to match the increased number of myeloid progenitors observed in organisms with a _JAK2_ mutation. To explore possible pathways downstream of _JAK2_ that could explain this link to myeloid diseases, we applied a machine learning approach (XGBoost) to AML TCGA data as a relevant and closely related blood cancer. We found that _JAK2_ is highly correlated with the NOTCH pathway (Fig. 3d), which has been found to act as a tumour suppressor in leukaemia due to the large expansion of GMP cells after loss of NOTCH signalling[35]. From the SHAP scores (SHapley Additive exPlanations, which are feature contribution measurements) in the classification of _NOTCH_ genes plotted in Fig. 3e, we identified _ITCH_ to be among the top 5 genes with the highest mean SHAP scores in the NOTCH pathway and found a pathway linking _JAK2_ to _ITCH_ from a search of the literature. _ITCH_ controls the degradation of NOTCH[36] and is found to be induced by _JNK1_[37] from the _MAPK_ pathway which is a well-known downstream pathway of _JAK2_/_STAT5_[38]. We therefore suggest that the _JAK2_ path to GMP expansion could be MAPK and NOTCH pathway dependent.

Another interaction predicted by our network model is the inhibition of _MYB by RUNX1_. The CMP are found to be differentially expanded between _JAK2_ and _TET2_ first patients in Ortmann et al[13]. In our initial model, we included an inhibition interaction between _SPI1_ and _MYB_, our CMP expansion marker, a connection which has been observed experimentally[39]. This inhibition and the stable _SPI1_

# a

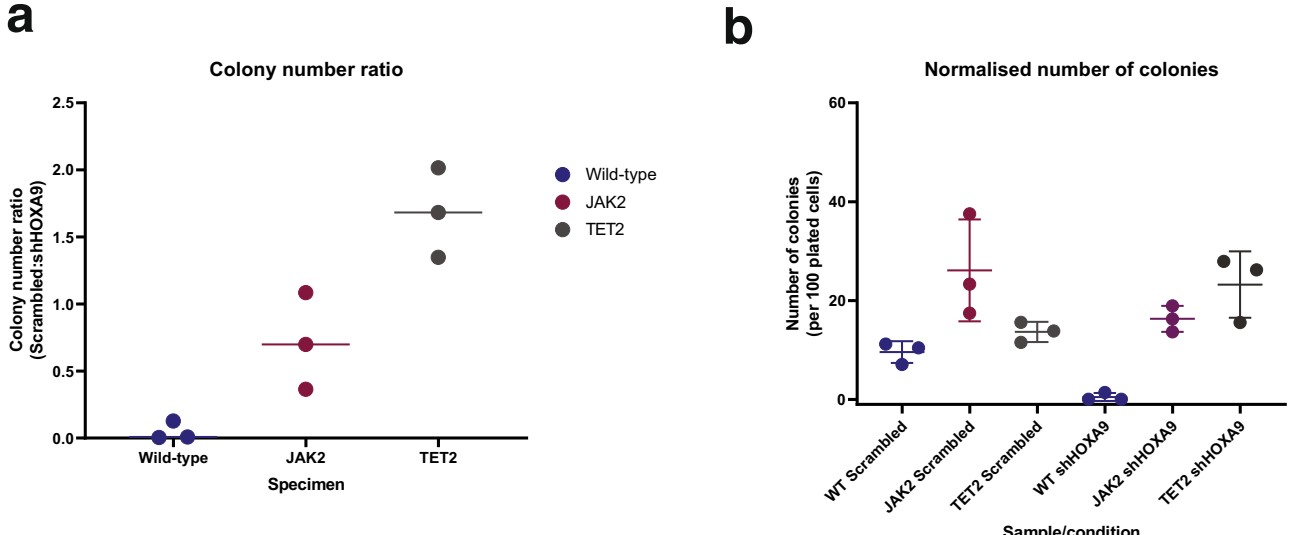

**Fig. 4 | *JAK2* activation of *HOXA9* improves colony survival in response to *HOXA9* inhibition. a** The ratios of colonies produced per sample are expressed as Scrambled:shHOXA9 colony number ratio with means. **b** The normalised colony count per sample and per condition with means and interquartile ranges of the three replicates. Two-way ANOVA statistical tests using multiple comparisons with FDR correction indicate a significant increase of colonies for *TET2* mutant versus WT cells with shHOXA9 ($q = 0.0032$) and *JAK2* mutant cells in both WT ($q = 0.0171$) and in shHOXA9 ($q = 0.0171$) contexts. Source data are provided as a Source Data file.

expression in the double mutant states prevented the known bifurcation in CMP expansion in double mutants. Further investigations lead us to suggest that the bifurcation could be obtained by replacing *SPI1* by *RUNX1* for *MYB* inhibition, and this additional set of interactions is supported by multiple studies. *RUNX1* activates *SPI1* and *GATA1*, and both are found to be inhibitors of *MYB*[39,40]. Additionally, conditional knockout of *RUNX1* in mice results in enhanced CMP frequencies[41,42] All together, these findings suggest that *RUNX1* can be linked to CMP expansion via *MYB* inhibition.

## Validation of the role of NOTCH pathway, *HOXA9* bimodality and its link to prognosis, and the interaction of *JAK2* with *HOXA9* through public datasets and experiments

To validate the predictions arising from our MPN computational model, we compared our findings to public MPN data not used in the model construction. Chen et al. compare MPN with different *JAK2* and *TET2* mutational profiles using transcriptomic mouse data[43]. We compared the gene expression of pathways/gene subsets to those we have included in our model to determine if our model fits their data. We first find that the NOTCH pathway behaves as predicted (Fig. 3f). We also find that the trend in the expression of *RUNX1*, *MYC,* and *MYB* support our model (Fig. S11–13 and Table S2). *HOXA9* expression also showed a "switching" behaviour in this mouse model, that is the first *JAK2* mutation has locked *HOXA9* into a specific expression level and the second mutation in the *JAK2*-first double mutant cohort does not subsequently cause it to revert to wild-type. Confusingly however, low expression of *HOXA9* is associated with *JAK2* mutations and high expression with *TET2* mutations. To confirm this trend, we examine the expression of other HOX genes that are closely correlated to *HOXA9* and find the same pattern.

Jeong et al.[44] previously demonstrated the direct phosphorylation of *TET2* by *JAK2* in a combination of in vitro human/murine hematopoietic cell lines with erythroid characteristics. Once phosphorylated *TET2* activates *KLF1*, an important positive regulator of erythropoiesis[45]. In this context, loss of *TET2* implies reduced erythroid differentiation which is in agreement with our model. In the same study, the authors show in a murine cell line that *JAK2* mutation leads to *HOXA9* upregulation. These findings are consistent with our *JAK2/TET2/HOXA9* motif but disagree with Chen's microarray

experiments where *HOXA9* expression is lowered in *JAK2* single and double mutants (Fig. 3f). Given the downstream genes follow the expected expression, this raises the question of whether the activations in the original motif should be replaced by a pair of inhibitions, to make the model consistent with Chen data. Whilst there exist possible routes to connect *TET2* and *HOXA9* through an inhibition, we are however unable to find evidence of inhibition of *HOXA9* by *JAK2*. We further note that as the Jeong data are human derived, it may be a more representative experimental model system. Future work using experiments in human samples could resolve this discrepancy. Both datasets however support the role of *HOXA9* as a binary switch in MPN.

In light of these observations, we sought to test the relationship between *JAK2* and *HOXA9* through experimentation. Our model predicts that mutation of *JAK2* would lead to activation of *HOXA9* through *STAT5*. *HOXA9* activity has been linked to cell viability[18] and inhibition of *HOXA9* would be expected to lead to a reduction of the number of colonies formed in a plating assay. We would therefore predict that mutation of *JAK2* would increase colony formation relative to wild-type when *HOXA9* is inhibited. Using wild-type and *JAK2* mutated stem and progenitor cells, we knocked down *HOXA9* and observed a reduction in colony formation in wild-type cells. We do, however, see a slight significant increase in colony formation in *JAK2*-mutant cells relative to wild-type whether or not *HOXA9* is inhibited ($q = 0.0171$). This finding is more consistent with our model, and the data set from Jeong et al.[44], where *JAK2* activates *HOXA9*. *TET2* behaviour is more complex, making an unsignificant increase to viability in WT but apparently synergistically increasing survival when *HOXA9* is inhibited ($q = 0.0032$, Fig. 4a, b). This suggests that colony formation is determined through complex *HOXA9* and *TET2* interactions, necessitating further study.

## Discussion

Out of 6817 genes tested *HOXA9* is the single most highly correlating factor for poor prognosis due to treatment failure in AML[17]. *HOXA9* could be argued to influence clinical characteristics in a continuous way, for example, if there was a broad unimodal distribution of *HOXA9* expression across patients and if *HOXA9* expression correlated with survival. Here we have demonstrated that instead it acts in AML as a discrete switch rather than a spectrum. This impacts AML clinical characteristics such as classification and survival. We further propose

the prognosis marker role of the *HOXA9* gene to another blood disorder, MPN. While *HOXA9* loss and overexpression are detrimental for normal cell development[16], our model assumes an intermediate activity for *HOXA9* in the healthy state. We show that in MPN diseases with *JAK2/TET2* mutations, *HOXA9* high expression is found in the *JAK2* first patients while *TET2* first patients display lower *HOXA9* expression. As *JAK2* first patients have a higher risk of developing thrombosis compared to *TET2* first patients, and as thrombotic events are the main causes of death in MPNs patients[46], this further suggests a deleterious influence of *HOXA9* high expression on patient clinical outcomes in another myeloid disease and emphasise the role of *HOXA9* as a marker of poor prognosis in blood malignancies.

In addition to providing insights into the regulatory control of cancer cell fate through *HOXA9*, our computational network model recapitulates the disease symptoms using well-known hematopoietic transcription factors. Further investigations of these genes could benefit clinicians by designing new drugs or applying already existing treatments to reduce symptoms and the risk of developing blast phase MPN. In addition to the specific claims of the model, several other clinical implications arise. Whilst *JAK2* is the main driver mutation found in all MPN patients, different diseases with distinct clinical traits can be observed[26]. Until now, the source of this clinical diversity following *JAK2* mutation was unclear. Here, we demonstrate that patients who first had a *TET2* mutation have a reduced number of erythroid cells as a result of *TET2* indirect downregulation of *GATA1* and *KLF1*, which explains the reduced number of PV diseases in *TET2* first patients despite the presence of *JAK2* mutation[13]. While *JAK2* dysregulation may be the principal driver of MPNs, other mutations shape the disease clinical type by altering the normal development of distinct hematopoietic subpopulations. Finally, we predict the involvement of the NOTCH pathway in MPN diseases. NOTCH shows both oncogenic and tumour suppressor roles in different tissues and in the hematopoietic system: NOTCH favours cancer growth in T acute lymphoblastic leukaemia through its *MYC* activation but is also found to augment the host immune response against cancer by activation of M1 macrophages[47]. The role of NOTCH in hematopoietic stem and progenitor cells is still an on-going debate, however, it seems that a certain level of *NOTCH* signalling is required to protect individuals from haematological malignancies[48]. We suggest that *JAK2* increases GMP expansion through its inhibitory effect on NOTCH via the MAPK pathway and *ITCH* and so predict a tumour suppressor role for NOTCH in the GMP cell population.

In building these models, several choices were made that potentially limit the further interpretation. Firstly, whilst all gene interactions included in this model are derived from studies of blood, due to paucity of information individual interactions may come from either mouse and human studies. Secondly, the precise role of the self-loop on *HOXA9* cannot be determined from our model alone. In the double mutant, the loss of the self-loop can lead to either abrogation (Fig. S6), wild-type (Fig. S7) or overexpression of *HOXA9* as a result of JAK2 constitutive activity (Fig. S5). This wild-type scenario in which *JAK2* and *TET2* mutations balance out *HOXA9* activity is able to respond to alternative orders of mutations through interactions between *SPI1* and *GATA1* (Fig. S12), albeit with phenotypes inconsistent with the disease[13]. Removing the *HOXA9* positive feedback loop in our model leads to its overexpression and loss of the bifurcation in the double mutant, changing *HOXA9* function in our model to obtain a wild-type expression restores the bifurcation in these cells (Fig. S12). However, CMP expansion is stable in both steady states which is not observed in patients with both mutations[13]. Finally, our model uses discrete values for gene expression and genomic data for validation, and represents a population of blood cells. Despite the historic successes of such approaches[49] modelling this network with continuous methods could help validate the model and give additional insights. Future work could also include using asynchronous updates and modelling the decision-making processes of individual cells, to better understand cancer fate commitment.

Our network model suggests a mechanism for understanding how cancer fate can be determined through regulatory switches and highlights several new areas for further studies. It also allows us to identify potentially important discrepancies in experimental studies.

## Methods

Our research complies with all relevant ethical regulations. The mouse study was undertaken under UK Home Office Licence granted to Dr. Kent (PEAD116C1) which was approved by the local AWERB committee and UK Home Office.

### Analysis and visualisation of public cancer datasets
AML patient data contains RNA sequencing information from 173 patients. We used the logarithmic Transcripts Per Million (log2 TPM + 1) normalised data. Low expressed genes are excluded (defined as a gene for which more than 50 samples have a TPM value <1). The R package multimode[50] was used to determine the significance of gene expression bimodality and the *modetest* command to reject unimodality with the default ACR (Ameijeiras-Crujeiras-Rodríguez) method, a multimodality test combining the use of a critical bandwidth and an excess mass statistic[51], using a *p* value of 0.05. We used the R package Survival[52] to plot the survival curves and compute the *p* values of the log-rank test. We plotted Sankey diagrams with the Plotly Python Open Source Graphing library (available at https://plot.ly).

### Differentially expressed genes in HOXA9 cohorts
We used a python script to separate patients between the two *HOXA9* expression peaks found in the AML data from TCGA. In all, 40 patients are found in the low peak (Fig. 2a), in which we remove the nine patients with a null value for *HOXA9* expression. We defined the high peak as the 80 patients with an expression between 4 and 5.5 for *HOXA9*. We found that subsequent analyses are robust to alternative high peak thresholds (Fig. S1). We compute the absolute difference of the mean expression of each gene between each cohort to find the genes which are most differentially expressed between the two groups of patients. We subsequently ranked the genes from the highest to the lowest absolute difference and take the top 30 genes from this list. This workflow was repeated using the fold change between cohorts. The top 30 genes from this set predominantly included either genes in the HOX family or genes with no determined role in haematopoiesis. This finding coincides with subsequent analyses of *HOXA9* cohorts using the R package DESeq2[53] in which differentially expressed genes cannot be classified into specific hematopoietic functions (Fig. S2).

### Microarray data analysis
While 12 samples are described in the paper we used for the MPN mouse transcriptomic data[43], only 11 could be found in the public data, with one wild-type sample missing in the microarray.

For analysis, from the set of all transcripts in the microarray, the genes with a low detection *p* value (below 0.05) were filtered and transformed with quantile normalisation. The ComplexHeatmap R library was used to plot the heatmaps[54].

### XGBoost
XGBoost (eXtreme Gradient Boosting) was used to rank different gene pathways that have been well described in cancer to identify which pathways and genes amongst these pathways have the highest correlation with *JAK2* and its expression level in the AML patients[55]. Thirteen pathways were chosen through the literature (Table S4). A model is trained and validated for each pathway. More details can be found in the Supplementary Information.

### Executable network model of MPN
Computational models of MPN cancer fate determination were constructed as a qualitative network (QN) in the BioModelAnalyzer[56]. This

process is described in more detail in Supplemental methods, but briefly QNs are constructed from reported gene interactions in the wider literature, and refined by testing model behaviour against reported phenotypes.

## Experimental validation of *JAK2*/*HOXA9* interaction

Full details are presented in Supplementary Methods. Briefly, all mice are originally on a C57/Bl6 background with the TET2 mice originally obtained via Prof. Anjana Rao (La Jolla, USA) and the JAK2 V617F mice obtained via Prof. Anthony Green (Cambridge, UK). Haemopoetic stem cells were isolated by flow cytometry cell sorting and cultured (Fig. S14, S15). For *HOXA9* gene knockdown experiments, three biological replicates were generated from each genotype for two different conditions (noneffective scrambled control- and shHOXA9-transduced cells). Colony forming assays were performed, with colonies characterised and counted after 14 days. Normalised number of colonies grown in each replicate was calculated per 100 colonies plated into each well. Statistical analysis to determine statistically significant differences was done through an unpaired Student's $t$ test (GraphPad Prism, v 9.0.2).

## Reporting summary

Further information on research design is available in the Nature Research Reporting Summary linked to this article.

## Data availability

The AML patient data were generated by TCGA and downloaded with Firebrowse (RNAseq, [http://firebrowse.org/]). The AML clinical data from TCGA was downloaded with cBioportal (www.cbioportal.com). The microarray dataset reported in ref. 43 is available in the ArrayExpress repository at European Molecular Biology Laboratory–European Bioinformatics Institute (http://www.ebi.ac.uk/arrayexpress/) and is accessible through the ArrayExpress accession number E-MTAB-2986. Raw colony count data are presented in the paper in full - images of colonies are available in Fig. S16. Source data are provided with this paper.

## Code availability

Python and R scripts described in this section are available at ref. 57.

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

## Acknowledgements

We thank members of the Hall group, the Fisher group at the University College London Cancer Institute, and Aleksandra Watson at the University of Cambridge for valuable discussions. B.A.H. acknowledges support from the Royal Society (grant no. UF130039), the Medical Research Council (grant no. MR/S000216/1), and Microsoft Research. J.F. was supported by the National Institute for Health Research University College London Hospitals Biomedical Research Centre and Cancer Research UK. L.C.C. is supported by a CR-UK Programme Foundation award to D.G.K. (DCRPGF\100008). The D.G.K. laboratory is supported by an ERC Starting Grant (ERC-2016-STG-715371), a CR-UK Programme Foundation award (DCRPGF\100008), and an MRC-AMED joint award (MR/V005502/1).

## Author contributions

L.T. performed the experiments, analysed data, and wrote all versions of the manuscript. B.H. conceived, supervised the study, and wrote and edited the manuscript. D.S. supported data analysis. D.K. and L.C. performed experimental validation. L.T., M.C., D.S., L.C., D.K., J.F., and B.H. all edited the manuscript.

## Competing interests

The authors declare no competing interests.
