## [Peer review file · Nature Communications]

REVIEWER COMMENTS

Reviewer #1 (Remarks to the Author):

The authors have made substantial revisions to their manuscript to address my comment from the initial. They have addressed all my concerns and the resulting paper is exciting and very high quality.

During copy editing please ensure a couple of visual changes are made:

- figures are all legible when printed as some fonts are still a little small (2D pathway names, 2C axis labels,
- The axis range on the new figures 2C and S10 should be far smaller as the model can only take the values 0,1 or 2 so having a range from -6 to 12 makes substantial changes from 1 to 2 seem insignificant.

- Simon Mitchell

Reviewer #2 (Remarks to the Author):

Edits made by the authors in response to the reviewers comments have improved the clarity of the manuscript, although some important aspects require further clarification. The addition of experimental detail in the methods section of the revised manuscript and the now available source code enable proper review of the results and conclusions, and raise a number of additional queries. Additional computer models on altered network structures are very useful to understand the role of feedback, however, in vitro experimental evidence for HOXA9 as the definitive memory module of TET2/JAK2 mutation order would be desirable. Nevertheless, I believe that a revised manuscript by Talarmin et al. will be a valuable contribution to the field of myeloid blood cell disorders that could be of significant interest to the readers of Nature Communications.

--

Methods and Figure 1a: The additional details are helpful to understand the stratification. In my opinion, this could be made even clearer through the following modifications:

- Indicate the selection thresholds in the histogram as vertical lines, or by colouring the low/high selection range.
- Since the first bin also contains patients with 0 HOXA9 expression (which are subsequently filtered) adding a rug plot to the histogram (or replacing the histogram with a violin plot) may facilitate distinction of the filtered data.

The addition of Figure S1 is helpful to understand the consequence of the thresholding, but I feel like a corresponding plot for the survival curve is also needed to understand why these particular thresholds were chosen. Alternatively, I would propose to simplify the thresholds - why exclude patients (barring them from proper prognosis) without justification (e.g. those with 0 expression: are they not HOXA9 low? Can you demonstrate that these are sequencing errors as stated in the response to referees? - those with expression > 5.5: are they not HOXA9 high?)?

--

The authors collate evidence from existing data and the literature that HOXA9 may serve as a central memory module for the JAK2/TET2 mutation sequence. However, this evidence is circumstantial and no ultimate proof in form of an in vitro experiment is provided. The computer model may be regarded as an experiment, but it is not clear if not other factors can replace HOXA9 (in particular since regulation of HOXA9 by JAK2 is indirect, and since the overall regulatory control of cell output is complex [see Fig.2b]). I therefore suggest to tone down both title and bold statements regarding the definitive function of HOXA9 throughout the manuscript. A new title (reflecting the unproven role of HOXA9) could be: 'HOXA9 may serve as a sequential logic gate, integrating the temporal order of TET2 and JAK2 mutations with implications for AML and MPN'.

With reference to the above, bimodality in HOXA9 is used as evidence that HOXA9 can act as a memory module, what other factors display bimodality? Are subgroups for these factors also associated with different survival?

--

Minor points:

- Figure 2D: Thank you for clarifying the MCC and error bars. The error bars now have been cropped at 100%? If this is the case, they no longer represent the standard deviation. I would recommend to re-plot the individual values of the various runs as points (instead of error bars), and to add a brief explanation to the figure legend where the individual points are coming from (similar to the response to referee 2).

- I do not agree with the use of 'switching behaviour' or 'switching property' to describe the constant expression of HOXA9 (lines 448, 510, 926). The 'self-sustaining property of the positive feedback' may be more appropriate here.

- Supplement has an unnamed TOC element (line 14, referencing page 16).

- Figure S3: What comparison does the p-value refer to? What statistical test was conducted?

- Figure S2: Please indicate in the figure legend, if positive log₂ fold-change values correspond to higher expression in 'HOXA9 high'. It would be helpful to know where HOXA9 itself is in this figure?

- The addition of 'continuous' in line 548 does not improve clarity; why not include more details from the response to referee 2 to explain this point?

- Line 329: [...] one mutation must activate the gene whilst the other inhibits it, [...]

Line 349: HOXA9 is therefore activated by both JAK2 and TET2 [...]

No evidence for inhibition of HOXA9 by TET2, but of reduced activation due to loss-of-function in TET2.

Reviewer #3 (Remarks to the Author): Expert in MPN molecular biology and translational research

This is quite an interesting story by Talarmain et al describing important and timely insight into the pathogenesis of post-MPN AML, a disease with a dismal survival rate and no treatment. There is a dearth of molecular understanding of what causes some MPNs to transform to AML, hence the lack of treatment options for these patients. This work provides new potential therapeutic avenues.

Overall, I find this work thorough, clear, and interesting. I can see that the manuscript has been greatly improved over its previous version.

My only comments are as follows: I appreciate that this is a computational paper, but do feel that some biological validation could greatly strengthen the manuscript. Because the biological problem the

authors are addressing is such an important one, a few simple experiments to validate their findings are critical. For instance, the authors may use a simple cell based assay to show that any number of their loss of HOXA9 models is correct, by using stem and progenitor cell behavior as a readout (i.e. in a colony forming / replating assay). I don't expect the authors to functionally validate all of their models, but even seeing one functionally validated would provide more certainty to the work.

The other suggestion I'd make is to include a simple model/graphical abstract in the supplemental figures, since it did take me some time to understand the model the authors were proposing!

Response to Referees' comments

We thank all referees for accepting to review our updated manuscript and are thankful for their insightful comments. We are pleased that all of the referees saw the improvements made in this new manuscript and agreed that our work is a valuable contribution to the field. We address the reviewers' comments and concerns below.

Reviewer #1 (Remarks to the Author):

The authors have made substial revisions to their manuscript to address my comment from the initial. They have addressed all my concerns and the resulting paper is exciting and very high quality.

During copy editing please ensure a couple of visual changes are made:

- figures are all legible when printed as some fonts are still a little small (2D pathway names, 2C axis labels,*
- The axis range on the new figures 2C and S10 should be far smaller as the model can only take the values 0,1 or 2 so having a range from -6 to 12 makes substantial changes from 1 to 2 seem insignificant.*

- Simon Mitchell

We thank the reviewer for his comments and have updated the figures accordingly.

Reviewer #2 (Remarks to the Author):

Edits made by the authors in response to the reviewers comments have improved the clarity of the manuscript, although some important aspects require further clarification. The addition of experimental detail in the methods section of the revised manuscript and the now available source code enable proper review of the results and conclusions, and raise a number of additional queries. Additional computer models on altered network structures are very useful to understand the role of feedback, however, in vitro experimental evidence for HOXA9 as the definitive memory module of TET2/JAK2 mutation order would be desirable. Nevertheless, I believe that a revised manuscript by Talarmin et al. will be a valuable contribution to the field of myeloid blood cell disorders that could be of significant interest to the readers of Nature Communications.

--

Methods and Figure 1a: The additional details are helpful to understand the stratification. In my opinion, this could be made even clearer through the following modifications:

- Indicate the selection thresholds in the histogram as vertical lines, or by colouring the low/high selection range.*
- Since the first bin also contains patients with 0 HOXA9 expression (which are subsequently filtered) adding a rug plot to the histogram (or replacing the histogram with a violin plot) may facilitate distinction of the filtered data.*

The addition of Figure S1 is helpful to understand the consequence of the thresholding, but I feel like a corresponding plot for the survival curve is also needed to understand why these particular thresholds were chosen. Alternatively, I would propose to simplify the thresholds - why exclude patients (barring them from proper prognosis) without justification (e.g. those with 0 expression: are they not HOXA9 low? Can you demonstrate that these are sequencing errors as stated in the response to referees? - those with expression > 5.5: are they not HOXA9 high?)?

The reviewer raises an interesting point. In our earlier submission we omitted patients with apparent zero-expression of HOXA9 for several reasons. Firstly, there was no smooth distribution around zero, suggesting that the peak at zero was artefactual. Secondly, HOXA9 is required for leukemia cell survival (<https://ashpublications.org/blood/article/113/11/2375/109995/HOXA9-is-required-for-survival-in-human-MLL>) and so cancer cells with apparently zero HOXA9 expression would not be expected to be viable. Finally, others in the field have found that zero-valued expression can be caused by sampling or technical errors in the sequencing experiments (<https://www.ncbi.nlm.nih.gov/pmc/articles/PMC7568192/>).

To further explore the nature of this zero-expression cohort and validate our approach we have examined overall gene expression and expression of HOX genes that are found to be correlated with HOXA9. We find that other genes express normally, and HOXA status is either consistent or inconsistent with low expression of HOXA9 depending on the patients. We have further performed PCA of expression across the whole population, labelling cohorts high, low, and zero. We find that zero expression patients co-locate with both high and low expressors. Together we believe this justifies our exclusion of the zero-expression cohort as several patients with null HOXA9 expression seem to be the result of a technical or sampling errors. Moreover, as slight changes in the low and high peak boundary choice does not interfere with the results (Figure S1), we have therefore decided to ignore those patients in the rest of the analyses.

We have also updated Figure 1A with a rug plot below the histogram as well as vertical lines to indicate the low and high peaks. We believe the addition of the rug plot can help the readers to better understand our choice of cohort thresholds. Indeed, there are a very low number of patients above 5.5 plus a cluster of patients is found around 4.5 and 5. Therefore, some patients above 5.5 may be outliers. Moreover, we choose to reduce the thresholds for the high peak to decrease the difference in number of patients between the two peaks.

--

The authors collate evidence from existing data and the literature that HOXA9 may serve as a central memory module for the JAK2/TET2 mutation sequence. However, this evidence is circumstantial and no ultimate proof in form of an in vitro experiment is provided. The computer model may be regarded as an experiment, but it is not clear if not other factors can replace HOXA9 (in particular since regulation of HOXA9 by JAK2 is indirect, and since the overall regulatory control of cell output is complex [see Fig.2b]). I therefore suggest to tone down both title and bold statements regarding the definitive function of HOXA9 throughout the manuscript. A new title (reflecting the unproven role of HOXA9) could be: 'HOXA9 may serve as a sequential logic gate, integrating the temporal order of TET2 and JAK2 mutations with implications for AML and MPN'.

The reviewer raises an important point about the value of experimental validation here. To address this issue in our earlier drafts we reanalysed relevant publicly available data that was not used in the construction of the model, and this allowed us to test our model and draw out contradictions not previously reported in the literature. We accept however that we have offered no new experimental evidence for the role of HOXA9 in the manuscript.

To try and address this concern, we have engaged external collaborators to perform *in vitro* experiments knocking down HoxA9 in JAK2 and TET2 mutated stem and progenitor cells. Despite 9 months of effort here, including challenges with primary HSC transduction and limited double mutant mouse numbers, we were unable to generate a full set of validation data. That said, we can share with reviewers data where HOXA9 has been knocked in WT and JAK2 mutant cells (new Supp Fig 13) where knockdown of HOXA9 has a slight, although not statistically significant, increase in

colony numbers in JAK2 relative to WT mice in the limited sample numbers. A rise in survival would be consistent with the model, where JAK2 activates HOXA9 and therefore would be expected to improve survival. Attempts with TET2 and JAK2/TET2 knockout mice failed to grow substantial numbers of colonies due to low cell input and poor cell transduction in both scrambled and shRNA conditions - these experiments would likely require an additional 3-6 months to undertake and analyse.

We acknowledge that this experiment does not specifically validate the claim of HOXA9 as a sequential logic gate and have updated the title accordingly. We have also made a number of adjustments to the text to soften these conclusions (throughout text e.g. L329, L595).

With reference to the above, bimodality in HOXA9 is used as evidence that HOXA9 can act as a memory module, what other factors display bimodality? Are subgroups for these factors also associated with different survival?

We thank the reviewer for these interesting questions. In our AML dataset we observe correlated bimodality across several HOX genes, but HOXA9 is the only gene with known links to both JAK2 and TET2. We find another set of bimodal genes, which correlate or anti-correlate to the HOXA9 status, with no known links to JAK2 or TET2. The correlation with HOXA9 status confounds our survival analysis, preventing us from assessing their impact on survival. Finally, we find two genes with bimodal distributions that are not correlated to the HOXA9 status- APP and IGSF10. IGSF10 shows no clear correlation with HOXA9 status, and within HOXA9 cohorts has no clear link to survival. APP is partially correlated with HOXA9, where there exist cohorts of HOXA9/APP with expression profiles- high/high, high/low, and low/high but no cohort with low expression of both. Within just the low-expression peak of APP we find a statistically significant link with survival. Neither have known links to JAK2 or TET2, or HOXA9. We have made notes on this in the text (L258-268).

More generally, it has been shown in several studies that some genes with bimodal expression can cluster patients with different disease subtypes or disease prognoses. A well-known example is breast cancer where different low or high expression of ER, PR and ERBB2 show different clinical characteristics with different survival probabilities. But this is also true for other types of cancer such as ovarian cancer in which a recent study has shown that a group of bimodal genes were excellent prognostic targets (<https://www.sciencedirect.com/science/article/pii/S1525157812000451>).

--

Minor points:

- Figure 2D: Thank you for clarifying the MCC and error bars. The error bars now have been cropped at 100%? If this is the case, they no longer represent the standard deviation. I would recommend to re-plot the individual values of the various runs as points (instead of error bars), and to add a brief explanation to the figure legend where the individual points are coming from (similar to the response to referee 2).

We have updated the figure with boxplots and dots. The legend has also been extended with an explanation on the method used to obtain the dots. L996

- I do not agree with the use of 'switching behaviour' or 'switching property' to describe the constant

expression of HOXA9 (lines 448, 510, 926). The 'self-sustaining property of the positive feedback' may be more appropriate here.

We appreciate that this could create confusion. We originally decided to use "switch" as this has been widely used historically in both the logical modelling community and cell commitment studies. To avoid any potential issues, we have defined this word explicitly as suggested by the reviewer.
L122

- Supplement has an unnamed TOC element (line 14, referencing page 16).

We have corrected this.

- Figure S3: What comparison does the p-value refer to? What statistical test was conducted?

We compute the p-value of log-rank test which compare all four curves. This new information is added to the figure legend of Figure S3 in the supplementary information.

- Figure S2: Please indicate in the figure legend, if positive log₂ fold-change values correspond to higher expression in 'HOXA9 high'. It would be helpful to know where HOXA9 itself is in this figure?

We have updated Figure S2 with HOXA9 label and updated the figure legend.

- The addition of 'continuous' in line 548 does not improve clarity; why not include more details from the response to referee 2 to explain this point?

We have updated the conclusion with a more detailed explanation on this matter. L588

- Line 329: [...] one mutation must activate the gene whilst the other inhibits it, [...]

Line 349: HOXA9 is therefore activated by both JAK2 and TET2 [...]

No evidence for inhibition of HOXA9 by TET2, but of reduced activation due to loss-of-function in TET2.

We have updated the manuscript. L357 and 376

Reviewer #3 (Remarks to the Author): Expert in MPN molecular biology and translational research

This is quite an interesting story by Talarmain et al describing important and timely insight into the pathogenesis of post-MPN AML, a disease with a dismal survival rate and no treatment. There is a dearth of molecular understanding of what causes some MPNs to transform to AML, hence the lack of treatment options for these patients. This work provides new potential therapeutic avenues.

Overall, I find this work thorough, clear, and interesting. I can see that the manuscript has been greatly improved over its previous version.

My only comments are as follows: I appreciate that this is a computational paper, but do feel that some biological validation could greatly strengthen the manuscript. Because the biological problem the authors are addressing is such an important one, a few simple experiments to validate their findings are critical. For instance, the authors may use a simple cell based assay to show that any number of their loss of HOXA9 models is correct, by using stem and progenitor cell behavior as a readout (i.e. in a colony forming / replating assay). I don't expect the authors to functionally validate all of their models, but even seeing one functionally validated would provide more certainty to the work.

We thank the reviewer for their comments. To try and address this concern, we have engaged external collaborators to perform in vitro experiments knocking down HoxA9 in JAK2 and TET2 mutated stem and progenitor cells. Despite 9 months of effort here, including challenges with primary HSC transduction and limited double mutant mouse numbers, we were unable to generate a full set of validation data. That said, we can share with reviewers data where HoxA9 has been

knocked in WT and JAK2 mutant cells (new Supp Fig 13) where knockdown of HOXA9 has a slight, although not statistically significant, increase in colony numbers in JAK2 relative to WT mice in the limited sample numbers. A rise in survival would be consistent with the model, where JAK2 activates HOXA9 and therefore would be expected to improve survival. Attempts with TET2 and JAK2/TET2 knockout mice failed to grow substantial numbers of colonies due to low cell input and poor cell transduction in both scrambled and shRNA conditions - these experiments would likely require an additional 3-6 months to undertake and analyse.

We acknowledge that this experiment does not specifically validate the claim of HOXA9 as a sequential logic gate and have updated the title accordingly. We have also made a number of adjustments to the text to soften these conclusions (throughout text e.g. L329, L595).

The other suggestion I'd make is to include a simple model/graphical abstract in the supplemental figures, since it did take me some time to understand the model the authors were proposing!

We have included an abstract figure into the main text as we indeed thought this new figure could be a great addition to the manuscript.

REVIEWERS' COMMENTS

Reviewer #2 (Remarks to the Author):

In the revised manuscript, the authors have addressed all of my concerns and clarified the remaining open questions.

I would like to make one last suggestions to improve Figure 1:

Different (inverted) colors have been used to indicate thresholds in A and B (low is red in A, while high is red in B). I recommend to make colors consistent.

I believe that the revised manuscript will be of great interest to the readers, and a great addition to the portfolio, of Nature Communications, and I recommend publication of the manuscript.

Reviewer #3 (Remarks to the Author):

The authors have sufficiently addressed my concerns. The manuscript is much stronger in its revised form.

Response to Referees' comments

We thank all referees for accepting to review our updated manuscript. We are pleased that all of the referees saw the improvements made in this new manuscript and recommended it for publication. We address the reviewers' comments below.

REVIEWERS' COMMENTS

Reviewer #2 (Remarks to the Author):

In the revised manuscript, the authors have addressed all of my concerns and clarified the remaining open questions.

I would like to make one last suggestions to improve Figure 1:

Different (inverted) colors have been used to indicate thresholds in A and B (low is red in A, while high is red in B). I recommend to make colors consistent.

I believe that the revised manuscript will be of great interest to the readers, and a great addition to the portfolio, of Nature Communications, and I recommend publication of the manuscript.

We have updated Figure 1A to match colors in Figure 1B.

Reviewer #3 (Remarks to the Author):

The authors have sufficiently addressed my concerns. The manuscript is much stronger in its revised form.